# Over-Production of Therapeutic Growth Factors for Articular Cartilage Regeneration by Protein Production Platforms and Protein Packaging Cell Lines

**DOI:** 10.3390/biology9100330

**Published:** 2020-10-09

**Authors:** Ali Mobasheri, Heonsik Choi, Pablo Martín-Vasallo

**Affiliations:** 1Research Unit of Medical Imaging, Physics and Technology, Faculty of Medicine, University of Oulu, FI-90014 Oulu, Finland; 2Department of Regenerative Medicine, State Research Institute Centre for Innovative Medicine, LT-08406 Vilnius, Lithuania; 3Departments of Orthopedics, Rheumatology and Clinical Immunology, University Medical Center Utrecht, 3508 GA Utrecht, The Netherlands; 4Versus Arthritis Centre for Sport, Exercise and Osteoarthritis Research, Queen’s Medical Centre, Nottingham NG7 2UH, UK; 5Kolon TissueGene, Inc., Rockville, MD 20850, USA; heonsik1@kolon.com; 6Healthcare Research Institute, Kolon Advanced Research Center, Kolon Industries, Inc., Magok-dong, Gangseo-gu, Seoul 07793, Korea; 7UD of Biochemistry and Molecular Biology, Instituto de Tecnologías Biomédicas de Canarias, Universidad de La Laguna, San Cristóbal de La Laguna, 38071 Tenerife, Spain; pmartin@ull.edu.es

**Keywords:** osteoarthritis, articular cartilage, degeneration, regeneration, growth factor, protein production platform, protein packaging cell line, transforming growth factor β1 (TGF-β1), GP2-293 cells, TissueGene-C

## Abstract

**Simple Summary:**

Osteoarthritis (OA) is the most common form of arthritis across the world. Most of the existing drugs for OA treat the symptoms of pain and inflammation. There are no drugs that can dure the disease. There are a number of new treatments for OA including cell therapy and gene therapy. This articles outlines the concept behind TissueGene-C, a new biological drug for OA. This new treatment includes cartilage cells mixed with a genetically modified cell line called GP2-293, which is effectively a “drug factory”, over-producing the growth factors that are important for cartilage regeneration and changing the environment inside joints. The mixture is injected into the affected knee joint. These cells are designed to be short-lived and cannot reproduce. Therefore, after they have done their job, they die and are cleared by immune cells. This is a new and modern approach to treating OA and TissueGene-C is the prototype cell therapy for OA. In the future, it is entirely possible to combine different clones of genetically engineered cells like GP2-293 that have been designed to over-produce a growth factor or biological drug with cells from the cartilage endplate of the intervertebral disc to treat degeneration in the spine.

**Abstract:**

This review article focuses on the current state-of-the-art cellular and molecular biotechnology for the over-production of clinically relevant therapeutic and anabolic growth factors. We discuss how the currently available tools and emerging technologies can be used for the regenerative treatment of osteoarthritis (OA). Transfected protein packaging cell lines such as GP-293 cells may be used as “cellular factories” for large-scale production of therapeutic proteins and pro-anabolic growth factors, particularly in the context of cartilage regeneration. However, when irradiated with gamma or x-rays, these cells lose their capacity for replication, which makes them safe for use as a live cell component of intra-articular injections. This innovation is already here, in the form of TissueGene-C, a new biological drug that consists of normal allogeneic primary chondrocytes combined with transduced GP2-293 cells that overexpress the growth factor transforming growth factor β1 (TGF-β1). TissueGene-C has revolutionized the concept of cell therapy, allowing drug companies to develop live cells as biological drug delivery systems for direct intra-articular injection of growth factors whose half-lives are in the order of minutes. Therefore, in this paper, we discuss the potential for new innovations in regenerative medicine for degenerative diseases of synovial joints using mammalian protein production platforms, specifically protein packaging cell lines, for over-producing growth factors for cartilage tissue regeneration and give recent examples. Mammalian protein production platforms that incorporate protein packaging eukaryotic cell lines are superior to prokaryotic bacterial expression systems and are likely to have a significant impact on the development of new humanized biological growth factor therapies for treating focal cartilage defects and more generally for the treatment of degenerative joint diseases such as OA, especially when injected directly into the joint.

## 1. Introduction

Growth factors (GFs) are evolutionary-conserved proteins that enhance the growth, proliferation, migration, survival, and differentiation of a variety of cell types [1,2,3]. They have the capacity to regulate the specialized function and phenotype of cells, whether they are added directly to cells or co-cultured with cells that have been engineered to over-express them [4]. GFs can stimulate proliferation in many cell types but there are a number of cell types, including mature neurons, that are postmitotic and cannot re-enter the cell cycle. Therefore, precursors and progenitors of more specialized cells can be stimulated with GFs to stimulate proliferation and differentiation [5]. GFs and their receptors can be grouped into “families”, based upon shared features of amino acid sequence, and into “superfamilies”, based upon shared structural folds [6,7,8]. Many GF families display significant evolutionary conservation in sequence; for example, homologs of the fibroblast growth factor (FGF), epidermal growth factor (EGF), and transforming growth factor β (TGF-β) families can be found across the animal kingdom, playing important roles in growth, tissue remodeling, and repair [9,10]. However, higher vertebrates have larger GF families than invertebrates. For example, there are currently 22 members of the FGF gene family in the human genome, but only one in that of *Drosophila melanogaster* and *Caenorhabditis elegans* [11].

GFs are relatively small and stable polypeptides that are secreted by cells in the body [12]. GFs are present in the extracellular matrix (ECM) as secreted or membrane-bound proteins [13]. GFs can regulate a variety of cellular behaviors including growth, migration, differentiation, apoptosis, and survival, in both positive and negative manners, in the context of homeostasis and neoplasia [14,15,16]. GFs produced by stem cells have an array of functions during development, and play important roles in the maintenance of tissue homeostasis and wound healing in adult skin [17] and in other connective tissues such as articular cartilage [18]. IGF-I and basic FGF have been shown to augment articular cartilage repair in vivo [18].

The transforming growth factor-β (TGF-β) superfamily is encoded by 33 genes and includes TGF-β, bone morphogenetic proteins (BMPs), and activins [19,20,21,22]. Recent evidence suggests that TGFs, BMPs, and activins have important roles in regulating immune responses in the context of infection, inflammation, and cancer [23,24,25]. TGF-β1 is the prototype member of the TGF-β family of growth and differentiation factors [26]. It is the best-studied factor among the TGF-β family proteins, with its diversity of roles in the control of cell proliferation and differentiation, wound healing, and immunoregulation, and key roles in pathology, for example, in skeletal diseases, fibrosis, and cancer [26]. In the synovial joint TGF-β1 is a pleiotropic cytokine that is important for the regulation of tissue homeostasis, degeneration, and regeneration [27,28,29,30]. Its action on articular cartilage is particularly dependent upon the context in which it acts, eliciting seemingly opposite effects under different experimental conditions; it may counteract pathological changes in a young healthy joint, altering its signaling during aging, and may be an active participant in pathology in OA joints [30]. In the context of the present review, the promotion of TGF-β1 activity in articular cartilage and inhibition of TGF-β1 activity in subchondral bone may provide new avenues of treatment for OA [31].

GFs can be produced by genetic engineering in the research laboratory setting, and exploited using biotechnology platforms for further applications and used in various clinical, therapeutic, and regenerative contexts [32,33]. In this paper, we focus on GFs for cartilage regeneration. We review the current state-of-the-art cellular and molecular biotechnology for the over-production of clinically relevant therapeutic proteins. We propose that transfected and irradiated protein packaging eukaryotic cell lines may be used as “cellular factories” for the over-production of therapeutic proteins and pro-anabolic growth factors, particularly in the context of regenerative medicine for treating focal cartilage defects and degenerative diseases of the joints, such as osteoarthritis (OA). These cellular tools may be used to produce cocktails and combinations of GFs for intra-articular injection in a concentrated format or even injected into synovial joints, combined with allogeneic primary cells (i.e., isolated chondrocytes), chondroprogenitors, or stem cells.

Mesenchymal stem cells (MSCs), were recently renamed as “medicinal signaling cells” by Arnold Caplan, the scientist who originally named them mesenchymal stem cells [34,35,36]. The cells were originally and officially named “mesenchymal stem cells” more than 25 years ago because it was then believed that these cells represented a unique class of cells from bone marrow and periosteum that could be isolated and expanded in culture while maintaining their multipotent in vitro capacity to be induced by biochemicals, growth factors, and environmental cues to form a variety of mesodermal phenotypes and tissues. He proposed this name change because he believes that “medicinal signaling cells” more accurately reflected the fact that these cells have the functional capacity to migrate to and home in on sites of injury or disease where they are thought to secrete bioactive factors with immunomodulatory and trophic (regenerative) properties. The new terminology implies that these are not stem cells but rather cells that make therapeutic drugs in situ with medicinal properties [36].

In the future, it is entirely conceivable to combine different clones of live genetically engineered cells with chondrocytes, osteoblasts, or MSCs as viable protein factories producing different combinations of growth factors or pro-inflammatory cytokine antagonists for intra-articular injections.

## 2. Osteoarthritis (OA)

OA is a progressive and degenerative condition that causes load-bearing synovial joints to become painful and stiff [37]. According to the World Health Organization (WHO), OA is the most common type of arthritis affecting millions of people worldwide (https://www.who.int/medicines/areas/priority_medicines/Ch6_12Osteo.pdf). Although the main symptoms of OA are joint pain and stiffness, some patients also experience swelling (effusion), tenderness, and a grating or crackling sound when moving the affected joint. OA can occur in any joint but the disorder most commonly affects joints in the knees, hips, hands, and the spine. Although OA is primarily related to aging, it is, along with many other forms of chronic disease, also associated with a wide variety of modifiable and non-modifiable risk factors that include: overweight and obesity [38], sedentarism [39] and lack of physical exercise [40], genetic predisposition, bone deformities or reduced bone mineral density, occupational injuries, repeated stress and trauma in sport, certain metabolic and endocrine diseases and, importantly, the female gender, especially after menopause. The major risk factors for OA are summarized in Figure 1.

In terms of disease initiation and molecular pathogenesis, it is believed that there is a long and asymptomatic “molecular phase”. This is the phase during which the patients feel no symptoms but there are molecular alterations in cartilage and possibly also other joint tissues. The silent “molecular phase” is followed many years later by changes that are visible on a plain x-ray radiograph and the gradual appearance of clinical symptoms [41]. In addition to the primary risk factors of aging, obesity, gender, and genetics, other inciting risk factors for OA may include previous joint trauma or history of repetitive joint injuries or even the presence of metabolic syndrome and endocrine disease [42]. However, the disease is primarily biomechanical. There are biomechanical [43], inflammatory [44], and metabolic [45] factors that have been demonstrated to play dominant roles in the initiation and progression of OA.

## 3. Growth Factors and OA

GFs are important for the synthesis and maintenance of articular cartilage in vivo and in vitro [18,46,47,48]. The use of bioactive GFs is under consideration as a potential therapy to enhance the healing of chondral injuries and modify the arthritic disease process [49,50]. The most important growth factors that are relevant to cartilage homeostasis are summarized in Table 1.

## 4. Mammalian Protein Production Platforms

Mammalian cell lines derived from human, mouse, and hamster tissues (Figure 2) are excellent hosts for the production of complex recombinant proteins that require extensive folding, the assembly of multiple subunits and posttranslational modifications including N-glycosylation and many others. Over the past 20 years, the industrial demand for recombinant therapeutic proteins has significantly increased [67,68]. Mammalian protein production platforms and protein packaging cell lines have been extensively used to produce recombinant proteins [69,70]. For these reasons, such mammalian cells are widely used by the pharmaceutical and biotechnology industries for the large-scale production of recombinant proteins [71,72,73], which may include diagnostic and therapeutic proteins, peptides, antibodies, and antibody fragments [74]. Different mammalian cell platforms are used according to the quantity and quality of the desired product required and the platforms can be scaled according to yield requirements [75]. The most commonly used mammalian cell lines found in the research and industrial therapeutic protein production settings are Chinese hamster ovary cells (CHO) [76,77] and human embryonic kidney 293 cells (HEK-293) [78]. Some of these expression systems are transient, whereas others are stable.

### 4.1. Transient Expression Systems for Recombinant Proteins

Transient expression platforms for the production of mammalian proteins often use human HEK-293 or hamster CHO cells [79,80]. HEK-293 is a cell line derived from human embryonic kidney cells grown in tissue culture and is widely used in cell biology and biotechnology [81]. The cells were derived from a human embryonic kidney but the phenotypic origin of the cells is thought to be neuronal (https://www.hek293.com/). CHO cells are an epithelial cell line derived from the ovary of the Chinese hamster [77]. CHO cells are used in diverse biological and medical research applications [82]. CHO cells are also used commercially for the production of clinically relevant therapeutic proteins. Glycoengineering of CHO cells is an active and promising area of research focusing on enhanced glycosylation capabilities for highly glycosylated proteins [82]. However, many of these expression systems are “transient”, meaning that they can only be manipulated acutely to drive over-expression of a desired protein for a given period of time. Therefore, they enable the rapid production of milligram to gram quantities of protein on a flexible scale within a few weeks. The detailed description of the workflow involved in the development of such tools is beyond the scope of this review article but in essence, it involves the transfer of the gene of interest into an expression vector, over-production and purification of the recombinant protein, quantification of the protein, determination of protein integrity, and specific functional studies, if necessary [83]. However, there are industrial applications that require sustained and stable production, packaging, and secretion of proteins.

### 4.2. Stable CHO Cell Line Development

It is possible to develop stable cell lines that continuously produce a target protein of therapeutic value. Establishing highly productive clonal cell lines with constant productivity over two to three months of continuous culture is extremely challenging but possible, and has already been achieved [84]. Transfected CHO DG44 cells are often used as a model for this purpose. They are cultivated under several rounds of methotrexate selection [85]. Monoclonal CHO-derived cell lines may be generated by subcloning pools of the most productive cells, and clone stability is confirmed. For example, stable cell lines have been designed to produce recombinant monoclonal anti-tumor necrosis factor α (TNF-α) antibody [86]. Once the clone has been established, the cells may be frozen and archived or shipped and distributed to other locations and banked at multiple sites.

### 4.3. Mammalian Protein Production Platforms for Large Scale Production of Therapeutic Proteins

Mammalian protein production platforms have important advantages as eukaryotic expression systems and are currently being employed as indispensable “cellular factories” for the large-scale production of humanized therapeutic antibodies and proteins [87]. Mammalian cell expression systems can now support the large-scale production of proteins, especially of those of clinical relevance and human origin [88]. Over the last few decades, these platforms have gradually evolved and found new applications in biology, biotechnology, and medicine. Protein production platforms have had a profound impact in many areas of basic and applied research, and an increasing number of biological drugs and vaccine antigens are now recombinant mammalian proteins made using these tools [89]. Recombinant proteins and therapeutic monoclonal antibodies are produced in mammalian cell lines. Bacterial expression systems (i.e., *Escherichia coli*) can still be used and indeed they have been used for many therapeutic proteins. However, these prokaryotic systems do not possess the protein processing machinery that is present in eukaryotic cells. Using eukaryotic models can optimize therapeutic protein production by ensuring that proteins are properly folded, and all the necessary post-translational modifications are introduced. It is important to note that correct folding and post-translational modifications are essential for appropriate biological activity. Therefore, these modifications should be properly introduced in the most appropriate eukaryotic and “mammalian” cellular context. Mammalian cell expression systems and protein production platforms are important tools for producing complex biotherapeutic proteins [90]. As already mentioned, various different mammalian expression systems are also being used for protein and glycoprotein production and recent cellular engineering strategies have been developed to increase glycoprotein productivity [91], an important feature that bacterial expression systems do not possess. At present, many vaccine production pipelines utilize mammalian cell expression systems and protein production platforms.

## 5. Viral and Non-Viral Gene Therapy for OA

Most of the early experimental progress in the area of gene therapy for OA was made with gene transfer to the synovium, a tissue that is particularly amenable to genetic modification by a variety of gene vectors, using both in vivo and ex vivo protocols [92]. However, despite the importance of targeting inflammatory pathways in the synovium to treat the synovitis associated with OA, the main research priority has been regenerative therapy for joints the focus has specifically been on cartilage regeneration and this is also where most of the gene therapy has been focused [93,94,95,96]. The focus so far has been upon the transfer of genes whose products enhance the synthesis of the cartilage ECM, or inhibit its breakdown, although there is certainly room for finding novel and alternative targets, which may include cytoprotective factors and molecular chaperones. There are also numerous possibilities and opportunities for targeting multiple catabolic and anabolic pathways, and, upstream regulators of key catabolic switches in chondrocytes and synoviocytes.

Early work on non-viral gene delivery used cationic liposomes and non-liposomal lipid formulations for cartilage regeneration. Goomer et al. proposed tested, von-viral in vivo gene therapy for articular cartilage and tendon repair [97]. They were successfully able to transfect TGF-β1, parathyroid hormone-related protein (PTHrP), and another marker gene into primary perichondrium and articular chondrocyte in situ with efficiencies of over 70%. Goomer et al. also demonstrated the efficacy of expression in a rabbit model of osteochondral defect, and repair a canine model of intrasynovial flexor tendon injury [97].

Madry et al. used isolated lapine chondrocytes transfected with an expression plasmid vector carrying the *Photinus pyralis* luciferase gene by a lipid-mediated gene transfer method using the reagent FuGENE 6 [98]. In their pioneering work, they proposed that lipid-mediated gene transfer to primary chondrocytes within a gel suspension delivery system directly into osteochondral defects and the sustained expression of the transgene in vivo may facilitate cartilage repair and may provide alternative treatments for articular cartilage defects [98].

The same team of researchers pushed the boundaries even further by demonstrating that therapeutic growth factor gene delivery (with cDNA encoding the human insulin-like growth factor I (IGF-I) as an exemplar) using encapsulated and transplanted genetically modified chondrocytes may be applicable to sites of focal articular cartilage damage [66]. They also tested gene transfer of human fibroblast growth factor 2 (FGF-2) via transplantation of encapsulated genetically modified articular chondrocytes to show that overexpression of FGF-2 enhances the repair of cartilage defects via stimulation of chondrogenesis [99].

Developing this technology also allowed these authors to look at co-overexpression of IGF-I/FGF-2 for early repair of cartilage defects in vivo and testing the prospect of providing protection for healthy neighboring cartilage tissue [100]. This also raises the possibility of achieving synergistic analgesic effects by combining different genes in a combination therapy approach, as used in the treatment of cancer. The data provided by this team demonstrate that the combined delivery of genes encoding multiple therapeutic growth factors to cartilage defects may have clinical value for promoting cartilage repair in vivo. The protective effect of combined IGF-I/FGF-2 co-overexpression on the neighboring articular cartilage is certainly interesting in the context of protecting healthy cartilage from damage, for example in the high-impact sport context.

Recombinant adenovirus associated vectors may be used to directly transfer candidate gene sequences in human articular chondrocytes in situ, providing a potent tool to modulate the structure of OA cartilage. Although very few preclinical animal studies in OA models have been performed thus far, equine models of OA have been proposed for proof of concept studies in translational models [101]. Several gene therapy clinical trials have also been carried out in patients with end-stage knee OA based on the intraarticular injection of human juvenile allogeneic chondrocytes overexpressing a cDNA encoding TGF-β1 via retroviral vectors [102].

Targeting the synovium is another approach and it has already been tried but unfortunately, it has not been possible to build a convincing clinical case for targeting interleukin-1 (IL-1β) as a key mediator of cartilage loss in OA, as the clinical trials conducted by AbbVie for targeting IL-1β have produced generally disappointing results. Nevertheless, the therapeutic effects of IL-1β receptor antagonist (IL-1Ra) gene transfer have been confirmed in three different experimental models of OA [92] and it is likely that targeting IL-1β may still be a viable solution for targeting the more inflammatory phenotypes of human OA [103]. Gene therapy may be combined with cell therapy for developing innovative new treatments for OA.

## 6. Cell Therapy for OA

A detailed discussion of cell and stem cell therapy for OA is beyond the scope of this review. The readers are referred to a series of excellent research and review articles that cover this topic [104,105] including several comprehensive reviews from our own group [106] that discuss the potential for using primary chondrocytes [107] adipose, bone marrow, and synovial mesenchymal stromal cells (MSCs) [108,109,110], menstrual blood-derived stem cells [111], and induced pluripotent stem cells (iPSCs) [112]. The most important point to make is that MSCs derived from patients with advanced OA exhibit attenuated chondrogenic activity, suggesting that these cells may be poor candidates for cell-based therapies for OA [113].

## 7. Therapeutic Growth Factors for the Treatment of OA

As outlined in the previous section, a number of GFs have been proposed as novel biological agents for cartilage regeneration [114]. GFs represent a broad range of biologically active agents that are capable of activating and stimulating the growth and repair of damaged tissues as well as protecting cells from premature death [115]. They, therefore, offer a very promising avenue for both treatment and further study, especially in the context of OA. As previously stated, cartilage degradation and subsequent OA is more common in people aged over 50 years, but people of any age have a significantly increased risk of cartilage and joint damage that may lead to post-traumatic osteoarthritis after sports or other joint injuries. GF treatments offer potential benefits to prevent OA (especially at the earliest stages of disease pathogenesis) later in life, as well as being an immediate consideration after a sports injury, when the prevention of further damage is a priority.

## 8. GP2-293 Protein Packaging Cells in TissueGene-C

TissueGene-C, developed by Kolon is a unique cell and gene therapy specifically developed for the treatment of knee OA. TissueGene-C is undoubtedly the current state-of-the-art cell and gene therapy platform for the treatment of OA. There are no other products like it in any biological therapeutic pipeline. In this unique product, transfected and irradiated protein packaging cell lines are used as “cellular factories” for the production of therapeutic TGF-β1. TissueGene-C is a unique combination of cell and gene therapy targeting knee OA. The treatment strategy is simple and is achieved through a single intra-articular injection of joint-derived chondrocytes mixed with irradiated GP2-293 cells, a protein production platform derived from HEK293 cells. The general concept for TissueGene-C is presented in Figure 3 and Figure 4.

It is important to emphasize that the human GP2-293 cell line is one of two key components of TissueGene-C. These engineered cells perform the vital function of over-producing the crucially important growth factor TGF-β1. According to Kolon TissueGene, GP2-293 cells have been used throughout the entire developmental process for TissueGene-C, from the first production of the Master Cell Bank (MCB) to the development of the working cell bank and the final live cell and gene therapy product formulation. As highlighted earlier, GP2-293 cells are derived from the HEK 293-based retroviral packaging cell line which is a well-known and widely utilized cellular model for protein expression and purification. However, in most laboratories, HEK 293 cells and their GP2-293 counterparts have been used for research only, mainly for large-scale protein production and packaging. GP2-293 is a powerful cellular platform for the over-production of therapeutically relevant human proteins and can be further manipulated and exploited. In the TissueGene-C development pipeline, this is the first time that such a human protein production platform has been incorporated into a live cell drug that is injected into the joint in the context of OA treatment and cartilage regeneration. Effectively, the GP2-293 cells in TissueGene-C are a protein-producing tool and “cellular factory”. We have previously emphasized that native patient-derived primary chondrocytes do not have the capacity to over-produce growth factors such as TGF-β1 in the high quantities needed for effective cellular therapy and regenerative applications. The subtle point that many scientists and clinicians working in the field of OA treatment appear to have missed or neglected is that transduced and irradiated GP2-293 cells may be transformed and triploid cells, but they are not cancer cells since they have lost their capacity for proliferation through irradiation. The key point is that they cannot proliferate. Therefore, the live GP2-293 cells in TissueGene-C cannot survive in the synovial joint for more than a few days and they simply cannot proliferate since they have been irradiated. These cells are expected to carry out their transient function as radiation inactivated transfection models in vivo. They are protein packaging tools and “cellular factories” for the over-production of high quantities of therapeutic TGF-β1. Thus far, no drug-related serious side effects have been identified in any of the patients enrolled in the clinical trials (http://www.businesskorea.co.kr/news/articleView.html?idxno=32318). After the GP2-293 cells have carried out their TGF-β1 over-production duty, they are expected to die, and their remains should be cleared by joint resident macrophages through the normal process of phagocytosis (Figure 5).

The key message of this paper is to highlight the scientific basis and rationale for the use of mammalian cell transfection models in the development of TissueGene-C. This has not been clear in previous publications and this is why further clarification is necessary. There is very extensive and well-established literature on the use of HEK-293 cells and their GP2-293 derivatives as transfection models and cell culture models for protein over-production. However, it is important to note that the efficacy and safety of HEK-293 cells and their GP2-293 derivatives have not been extensively reviewed in regenerative medicine. However, the risks are low and manageable and the prospects for future use of transfection tools in regenerative medicine and cell therapy look bright and positive, especially since native and untransformed primary cells such as chondrocytes are not robust protein over-production platforms and do not have the regenerative and immunomodulatory capacity needed for reprogramming the micro-environment of the joint. Only a cellular factory and protein over-production platform that can transiently over-produce TGF-β1 and other growth factors belonging to the family of BMPs and its subfamily of growth differentiation factors (GDFs) has the potential to address the conceptual problems that researchers face in this area. The particularly the short half-life of growth factors such as TGF-β1 makes the purified protein by itself a feeble and ineffective tool, requiring a live cell platform capable of sustained over-production of the growth factor.

## 9. Conclusions

The concept of using gene therapy for cartilage repair originates from the idea of transferring genes encoding therapeutic growth factors into the joint to promote tissue repair and regeneration [116,117,118,119]. Delivering genes into a degenerate synovial joint, even temporarily, could induce changes in the inflammatory micro-environment. Whether the genes are targeted to the synovial space, the synovium or articular cartilage, the spatially defined delivery of therapeutic molecules to sites of cartilage damage could facilitate endogenous tissue repair and regeneration [116]. However, gene expression levels may be low, inefficient, or overwhelmed by the inflammatory and catabolic micro-environment. Genetically engineered cells can provide the “factories” for the transcription of genes encoding therapeutic growth factors. Conventionally, cells have been used to over-produce proteins and then the over-produced proteins have been used in a purified and concentrated form experimentally in therapeutic applications. However, it is also possible to use live cells to deliver a therapeutic protein in a continuous but time-limited manner. The ability to deliver large quantities of a therapeutic growth factor in a time-limited manner is important, especially for TGF-β1, because sustained production of this growth factor can drive osteophyte formation and synovial fibrosis in OA joints [120,121,122,123,124]. Although TGF-β1 acts as a protective, load-induced factor in a young healthy joint, its sustained presence can function as a deleterious factor in an OA joint [125,126]. The protective function of TGF-β1 is decreased in aged cartilage compared to young, and this is one of the reasons for using TGF-β1 as a therapeutic factor. Cell factories, such as GP-293 cells have the capacity to over-produce large quantities of this growth factor and if they have been previously irradiated and rendered incapable of replication, they then gradually decline in activity and die off within the synovial joint, thus avoiding any possibility of long-term concerns about osteophyte formation and synovial fibrosis. Importantly, irradiation of the cells also renders them incapable of replication and survival beyond a few days in the synovial space.

Therapeutic strategies that combine cell and gene therapy are now a reality with significant potential for clinical development in orthopedics and rheumatology [127,128]. However, cell and gene therapy already rely on protein production platforms and protein packaging cell lines as the necessary tools for the over-production of the desired quantity of therapeutic proteins. Future cell and gene therapy strategies for treating OA and promoting cartilage and intervertebral disc (IVD) regeneration may exploit the potential for using mammalian protein production platforms and irradiated and transfected protein packaging cell lines for over-production of therapeutic proteins and growth factors, individually or in combination, within the synovial joint or even in the IVD [129,130]. Further developments in this area may include combinations of cell clones that over-produce several growth factors and cell and gene therapies that may be used to target other joint tissues and the IVD in the spine (Figure 6) [130].

Most of the emerging biological agents in the current drug development pipelines are produced using mammalian protein production platforms [131]. Protein production tools are essential for large-scale production of therapeutic proteins and growth factors. These tools may be used to generate functional native and mutant proteins with appropriate folding, assembly, and posttranslational modifications. In most cases, these platforms have been used to produce therapeutic proteins that are then purified, characterized, and incorporated into products and derivatives. However, there is potential for using protein packaging cells in new therapies that include live cells for the transient release of large concentrations of growth factors with short half-lives directly into the joint or the spine. The concept for TissueGene-C has been reviewed in this paper as the current state of the art for the treatment of knee OA [129]. A study published in July 2020 investigated the efficacy and mechanism of action of TissueGene-C in a rat model of OA. Using the monosodium-iodoacetate (MIA) model of OA, we have demonstrated that TissueGene-C provides pain relief and cartilage structural improvement in the MIA OA model over 56 days. In parallel with these long-term effects, cytokine profiles obtained on day four revealed increased expression of interleukin-10 (IL-10), an anti-inflammatory cytokine, in the synovial lavage fluid. Moreover, the increased levels of TGF-β1 and IL-10 stimulated by TissueGene-C induced the expression of arginase 1, a marker of M2 macrophages, and decreased the expression of CD86, a marker of M1 macrophages. These novel results suggest that TissueGene-C exerts a beneficial effect on OA by inducing an M2 macrophage-dominant micro-environment [132].

The stark realization that many primary, aged and senescent chondrocytes and MSCs possess feeble regenerative properties means that future regenerative medicine and tissue engineering strategies for the joints and the spine could use primary allogeneic cells or stem cells combined with mammalian protein production platforms to drive the production of therapeutic proteins and pro-anabolic growth factors [129,130]. We now have the capacity to manipulate the replicative potential of the cellular component of protein production platforms by x-ray or gamma irradiation and make their replication incompetent. This will allow us to inject them directly into the joint in order to drive GF production in a controllable way. Advances in the field of molecular and cellular biotechnology and the development of new and more robust and customizable protein production platforms are likely to have a positive impact on the treatment of arthritic diseases, tissue engineering, and regenerative treatments for the musculoskeletal system.

## Figures and Tables

**Figure 1 biology-09-00330-f001:**
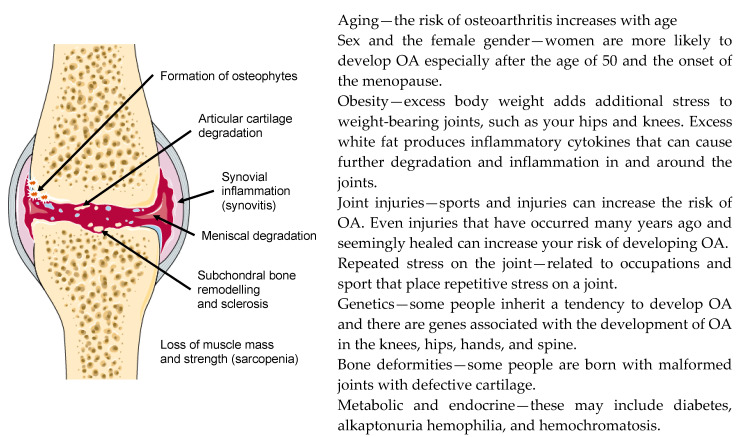
Major structural changes that occur in the joint (**left**) and risk factors for the development of osteoarthritis (OA) (**right**).

**Figure 2 biology-09-00330-f002:**
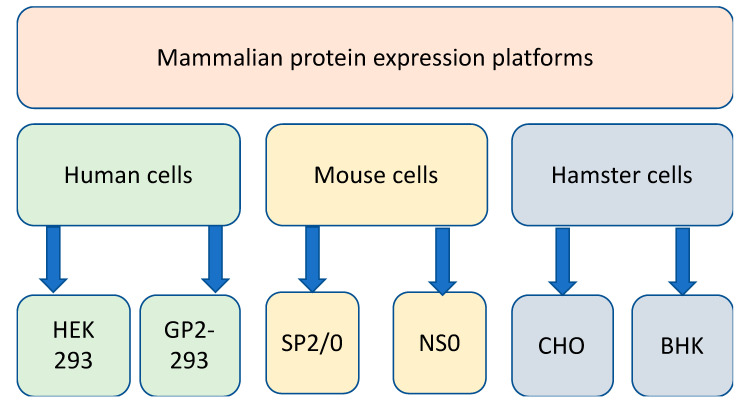
Mammalian protein production platforms using human, mouse, and hamster cell lines.

**Figure 3 biology-09-00330-f003:**
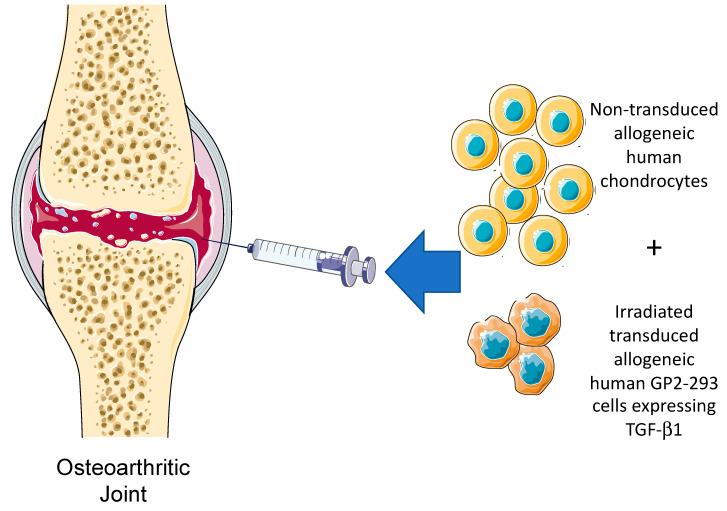
The intra-articular injection concept for TissueGene-C, a novel cell and gene therapy targeting knee OA through a single intra-articular injection of non-transduced allogeneic human chondrocytes, irradiated transduced human GP2-293 cells. These cells have been engineered to over-produce TGF-β1, the key biological growth factor with the ability to promote anabolic repair and regeneration and induce changes in the inflammatory milieu and microenvironment of the synovial joint.

**Figure 4 biology-09-00330-f004:**
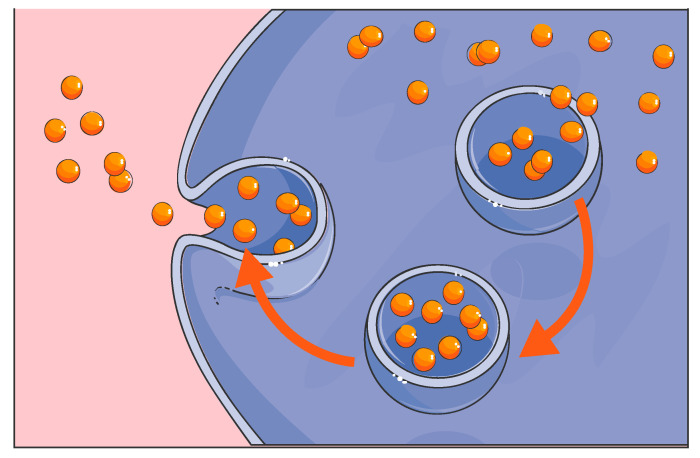
Production and secretion of therapeutic GFs by GP2-293 cells in TissueGene-C.

**Figure 5 biology-09-00330-f005:**
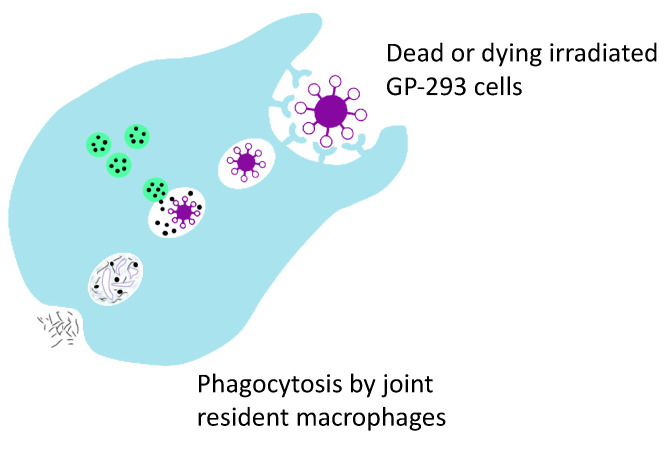
Phagocytosis is believed to be responsible for the clearance of dead GP2-293 cells and their debris by joint resident macrophages.

**Figure 6 biology-09-00330-f006:**
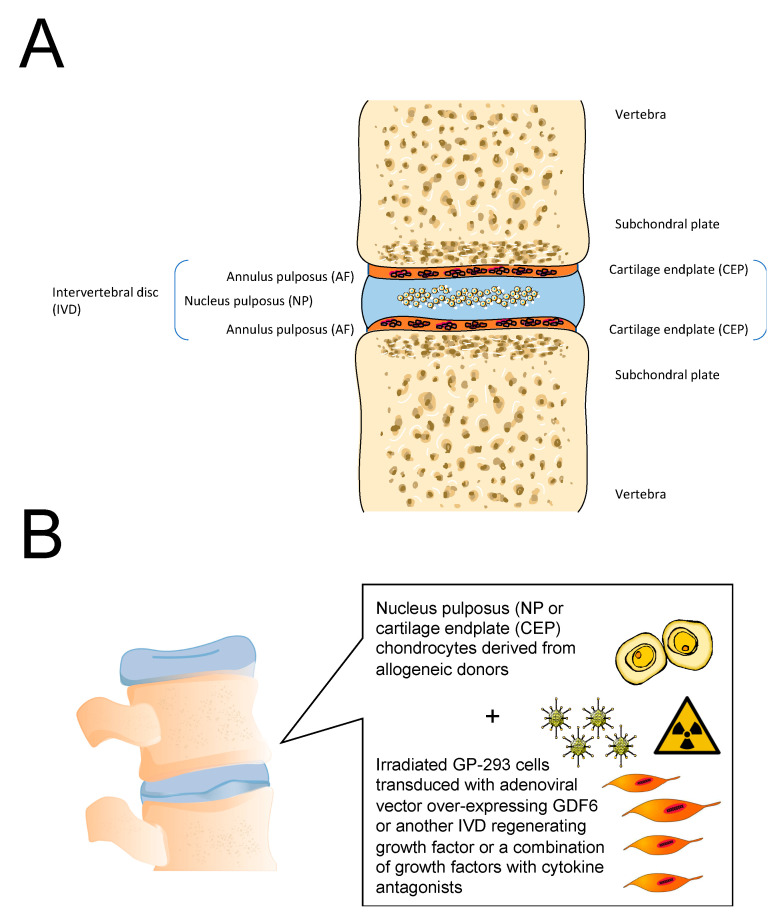
(**A**). Structure of the healthy intervertebral disc (IVD). (**B**). General concept for cell and gene therapy targeting the nucleus pulposus (NP) and the cartilage endplate (CEP) in the IVD.

**Table 1 biology-09-00330-t001:** Major growth factors involved in cartilage homeostasis, the development of osteoarthritis (OA), and applications in cartilage and bone repair and regeneration. Some of the growth factors listed have negative as well as positive impacts on joint tissues [51].

Growth Factor	Function	References
Platelet-derived growth factor (PDGF)	Regulates the secretion and synthesis of collagen	[50,52,53]
Epidermal growth factor (EGF)	Stimulates cellular proliferation, endothelial chemotaxis, and angiogenesis	[54,55]
Vascular endothelial growth factor (VEGF)	Increases angiogenesis and vascular permeability	[56]
Transforming growth factor-β1 (TGF-β1)	Stimulates the proliferation of undifferentiated mesenchymal stromal cells (MSCs), stimulates chemotaxis of endothelial cells and angiogenesis	[57]
Basic fibroblast growth factor (bFGF)	Promotes the growth and differentiation of chondrocytes and osteoblasts stimulates mitogenesis of mesenchymal cells, chondrocytes and osteoblasts	[58,59]
Connective tissue growth factor (CTGF)	Contributes to joint homeostasis and OA severity by controlling the matrix sequestration and activation of latent TGF-β1	[60,61]
Insulin-like growth factor 1 (IGF-1)	An important anabolic factor in cartilage homeostasis. IGF-1 promotes the synthesis of aggrecan, link protein, and hyaluronan, and inhibits proteoglycan degradation	[62,63,64,65,66]

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
