# Peer review of "Over-Production of Therapeutic Growth Factors for Articular Cartilage Regeneration by Protein Production Platforms and Protein Packaging Cell Lines"

_biology, 2020, doi:10.3390/biology9100330_

Round 1
Reviewer 1 Report
This is a review of protein packaging and secretion cell line systems with emphasis on TissueGene-C for insertion of growth factor TGFβ into synovial joints for osteoarthritis treatment.
While the authors refer to IGF-1 in the text, this "classic" growth factor is omitted from Table 1. Why?
The authors consider various methods besides Cell therapy to deliver proteins to tissues but omit discussion of direct supply of selected protein bound reversibly to a carrier matrix. This method could be especially important in osteoarthritis as cartilage matrix molecules and their derivatives are capable of binding growth factors.
Clarity is needed about where in the joint the product is placed. Injection into the synovial space is shown. Issues about diffusion of protein into articular cartilage matrix , the presence of growth factor inhibitors within the matrix are not discussed in the text but at least should be recognized.
Perhaps a table comparing advantages and disadvantages of various methods of inserting therapeutic proteins into synovial joints would be useful.
line 307
Native patient derived chondrocytes simply do not have the capacity to over-produce TGF-1 in sufficiently high quantities for effective cellular therapy and regenerative applications.
What is the evidence for this statement? This may apply to chondrocytes ex- vivo but under appropriate circumstances articular chondrocytes within articular cartilage particularly in osteoarthritis can replicate and produce new matrix.
Line 313
the 313 cells cannot survive for more than a very short period after being injected into the joint.
How long is a very short period? Is it sufficient to deliver adequate protein to stimulate regeneration?
Use of trade name 'TissueGene-C" in the text. This is a decision for Journal policy.
Author Response
Response to Reviewer 1
This is a review of protein packaging and secretion cell line systems with emphasis on TissueGene-C for insertion of growth factor TGFβ into synovial joints for osteoarthritis treatment.
While the authors refer to IGF-1 in the text, this "classic" growth factor is omitted from Table 1. Why?
Thank you for this suggestion. We have added IGF-1 in table 1.
The authors consider various methods besides Cell therapy to deliver proteins to tissues but omit discussion of direct supply of selected protein bound reversibly to a carrier matrix. This method could be especially important in osteoarthritis as cartilage matrix molecules and their derivatives are capable of binding growth factors.
Thank you for this important comment and observation. We have written another review article for Frontiers and in that paper, this issue has been explored. Therefore, we will not be discussing this matter in the revised manuscript.
Clarity is needed about where in the joint the product is placed. Injection into the synovial space is shown. Issues about diffusion of protein into articular cartilage matrix , the presence of growth factor inhibitors within the matrix are not discussed in the text but at least should be recognized.
Thank you for this comment but this information is already in the public domain. This particular product is injected directly into the synovial space, into the synovial fluid, not the articular cartilage or the synovium. In fact, ultrasound guidance is required to ensure that the product is not injected into cartilage or the synovium.
Perhaps a table comparing advantages and disadvantages of various methods of inserting therapeutic proteins into synovial joints would be useful.
Thank you for this important comment and observation. As stated earlier, we have already written and submitted another review article for Frontiers and in that paper, this issue has been explored. Therefore, we will not be discussing this matter in the revised manuscript.
line 307
Native patient derived chondrocytes simply do not have the capacity to over-produce TGF-1 in sufficiently high quantities for effective cellular therapy and regenerative applications.
What is the evidence for this statement? This may apply to chondrocytes ex- vivo but under appropriate circumstances articular chondrocytes within articular cartilage particularly in osteoarthritis can replicate and produce new matrix.
Chondrocytes undergo senescence and lose their capacity for producing growth factors. The fact this information is already in the public domain and relates to the comments that you made earlier about IGF-1. Senescent cells do not have the capacity to overproduce growth factors.
Line 313
the 313 cells cannot survive for more than a very short period after being injected into the joint.
How long is a very short period? Is it sufficient to deliver adequate protein to stimulate regeneration?
The cells do not survive on for more than a week to 10 days. This transient period of cell survival and protein expression appears to be sufficient for the therapeutic impact.
Use of trade name 'TissueGene-C" in the text. This is a decision for Journal policy.
Please note that the trade name Invossa is no longer used and has been changed to "TissueGene-C". This is the new trade name for the Kolon product.

Reviewer 2 Report
The review entitled “Over-production of therapeutic growth factors for 2 articular cartilage regeneration by protein production 3 platforms and protein packaging cell lines” is an interesting and well written manuscript. However, I have some issues that the authors should address.
Since the review is focused on growth factors for cartilage regeneration, I would suggest to improve section 3 and 7 of the review and summarize section 4.
Could the authors add a table describing the viral and non-Viral gene therapies in OA?
Figure 4 is composed by two identical carton. I suggest to use only one and to improve the figure legend.
Section 8: the authors discuss how works the TissueGene-C, developed by Kolon. In my opinion, this part is too biased and needs to be improved showing also that this system has been used in vivo (as reported in the conclusion).
Line 208-211: the sentence is too long.
Line 211: the comma should be changed to a full stop.
Author Response
Response to Reviewer 2
The review entitled “Over-production of therapeutic growth factors for 2 articular cartilage regeneration by protein production 3 platforms and protein packaging cell lines” is an interesting and well written manuscript. However, I have some issues that the authors should address.
Thank you very much for your constructive comments on our paper. We have addressed the issues that you have highlighted below.
Since the review is focused on growth factors for cartilage regeneration, I would suggest to improve section 3 and 7 of the review and summarize section 4.
Thanks for this suggestion. We have expanded Table 1 in section 3 and edited and re-written section 4 but since section 7 is not a key component of this review, we have not expanded it.
Could the authors add a table describing the viral and non-Viral gene therapies in OA?
Thanks for this suggestion. You make a very important point. However, the authors have done this for another paper that has been submitted to Frontiers. Therefore, we will not be including such a table in this particular paper.
Figure 4 is composed by two identical carton. I suggest to use only one and to improve the figure legend.
Thanks for this suggestion. We have done as you suggested.
Section 8: the authors discuss how works the TissueGene-C, developed by Kolon. In my opinion, this part is too biased and needs to be improved showing also that this system has been used in vivo (as reported in the conclusion).
We do not agree. We do not believe that this section is biased. This section simply outlines the key facts. We do not have any conflicts of interest to declare in relation to stating the key facts concerning this product and this is information that is to be in the public domain.
Line 208-211: the sentence is too long.
Thanks for this suggestion. This has been edited as you suggested. The sentence has been shortened.
Line 211: the comma should be changed to a full stop.
Thanks for this suggestion. This has been edited as you suggested.
